Investigating the bacterial community of gray mangroves (Avicennia marina) in coastal areas of Tabuk region

Ghabban Hanaa 1 2 h_ghabban@ut.edu.sa
Albalawi Doha A. 1 2
Al-otaibi Amenah S. 1 2
Alshehri Dikhnah 1 2
Alenzi Asma Massad 1 2
Alatawy Marfat 1 2
Alatawi Hanan Ali 3
Alnagar Dalia Kamal 4
Bahieldin Ahmad 5 6
1 Department of Biology, Faculty of Science, University of Tabuk , Tabuk , Saudi Arabia
2 Biodiversity Genomics Unit, Faculty of Science, University of Tabuk , Tabuk , Saudi Arabia
3 Department of Biological Sciences, University Collage of Haqel, University of Tabuk , Tabuk , Saudi Arabia
4 Department of Statistics, Faculty of Science, University of Tabuk , Tabuk , Saudi Arabia
5 Department of Biological Sciences, Faculty of Science, King Abdulaziz University , Jeddah , Saudi Arabia
6 Department of Genetics, Faculty of Agriculture, Ain Shams University , Cairo , Egypt
Souza Valeria
Electronic publication date: 2024 Oct 18
Publication date: 2024
Volume: 12
Electronic Location ID: e18282
Received 2024 Feb 28; Accepted 2024 Sep 19
Copyright: © 2024 Ghabban et al.
Copyright year: 2024
Copyright holder: Ghabban et al.
License: This is an open access article distributed under the terms of the Creative Commons Attribution License, which permits unrestricted use, distribution, reproduction and adaptation in any medium and for any purpose provided that it is properly attributed. For attribution, the original author(s), title, publication source (PeerJ) and either DOI or URL of the article must be cited.
License URL: https://creativecommons.org/licenses/by/4.0/

Keywords: Mangroves, Tabuk region, Avicennia marina, Microbiome, Soil, Bacterial diversity, Red sea, 16S rRNA

Funding: Deanship of Research and Graduate Studies at University of Tabuk S-0178-1443 This work was supported by the Deanship of Research and Graduate Studies at University of Tabuk through Research no. S-0178-1443. The funders had no role in study design, data collection and analysis, decision to publish, or preparation of the manuscript.

==============================
Mangrove vegetation, a threatened and unique inter-tidal ecosystem, harbours a complex and largely unexplored bacterial community crucial for nutrient cycling and the degradation of toxic pollutants in coastal areas. Despite its importance, the bacterial community composition of the gray mangrove (Avicennia marina) in the Red Sea coastal regions remains under-studied. This study aims to elucidate the structural and functional diversity of the microbiome in the bulk and rhizospheric soils associated with A. marina in the coastal areas of Ras Alshabaan-Umluj (Umluj) and Almunibrah-Al-Wajh (Al-Wajh) within the Tabuk region of Saudi Arabia. Amplicon sequencing targeting the 16S rRNA was performed using the metagenomic DNAs from the bulk and rhizospheric soil samples from Umluj and Al-Wajh. A total of 6,876 OTUs were recovered from all samples, of which 1,857 OTUs were common to all locations while the total number of OTUs unique to Al-wajh was higher (3,011 OTUs) than the total number of OTUs observed (1,324 OTUs) at Umluj site. Based on diversity indices, overall bacterial diversity was comparatively higher in rhizospheric soil samples of both sites. Comparing the diversity indices for the rhizosphere samples from the two sites revealed that the diversity was much higher in the rhizosphere samples from Al-Wajh as compared to those from Umluj. The most dominant genera in rhizosphere sample of Al-Wajh were Geminicoccus and Thermodesulfovibrio while the same habitat of the Umluj site was dominated by Propionibacterium, Corynebacterium and Staphylococcus. Bacterial functional potential prediction analyses showed that bacteria from two locations have almost similar patterns of functional genes including amino acids and carbohydrates metabolisms, sulfate reduction and C-1 compound metabolism and xenobiotics biodegradation. However, the rhizosphere samples of both sites harbour more genes involved in the utilization and assimilation of C-1 compounds. Our results reveal that bacterial communities inhabiting the rhizosphere of A. marina differed significantly from those in the bulk soil, suggesting a possible role of A. marina roots in shaping these bacterial communities. Additionally, not only vegetation but also geographical location appears to influence the overall bacterial composition at the two sites.

Introduction

Mangrove trees, a unique inter-tidal ecosystem accounting for 60–70% of tropical and sub-tropical coastlines are home to genetically diverse aquatic and terrestrial organisms. They are of great ecological importance as they are involved in protecting coastlines, influencing global climate patterns and facilitating phytoremediation processes in the coastal areas (Srikanth, Lum & Chen, 2016; Thatoi et al., 2013; Yu et al., 2020). Mangrove ecosystems are highly productive and biochemically dynamic, characterized by large reservoirs of organic matter, elevated salinity levels, increased rates of nutrient cycling, and regular tidal flooding (Holguin, Vazquez & Bashan, 2001; Palit et al., 2022; Thatoi et al., 2013). Due to high salinity in soils of coastal areas, plants growing in these regions have various salt tolerance strategies compared to terrestrial plants e.g., Avicennia marina, a gray mangrove also known as “pioneer of mangroves” harbours finger-like respiratory roots (pneumatophores) and leaves with salt glands as a combat strategy against higher salinity (Thatoi, Samantaray & Das, 2016; Waisel, Eshel & Agami, 1986).

Mangrove ecosystems are rich in microbial diversity, supporting the growth of various microbial communities such as bacteria, fungi, archaea, and protists, which play a crucial role in the cycling of essential nutrients like nitrogen, sulfur, and carbon, thereby maintaining soil chemistry (Alongi, Christoffersen & Tirendi, 1993; Palit et al., 2022). In these environments, bacteria and fungi dominate, accounting for 91% of the microbial diversity, while algae and protozoa represent only 7% and 2% respectively (Thatoi et al., 2013). The microbial communities in soils under mangrove vegetations are vital for the ecosystem’s maintenance, productivity and conservation, forming mutualistic relationships with mangrove plants as microbes living in roots produce phytohormones, and provide protection against phytopathogens, while receiving carbon metabolites from the plants in return (Durán et al., 2018; Holguin, Vazquez & Bashan, 2001; Liu et al., 2019; Sasse, Martinoia & Northen, 2018). Sulfate-reducing bacteria, methanogenic archaea, methylotrophic bacteria, nitrogen-fixing bacteria, and phosphate-solubilizing bacteria are the primary bacterial groups found in mangrove ecosystems (Alzubaidy et al., 2016; Purahong et al., 2019; Thatoi et al., 2013; Vazquez et al., 2000). Microbial communities inhabiting mangrove ecosystems are of great biotechnological significance as well, due to their possession of genes for producing beneficial antibiotics, enzymes, proteins, and salt-tolerance abilities (Xu et al., 2014).

Despite their significant ecological importance and biotechnological potential of the microbes they host, mangrove forests are globally threatened by deforestation, rising sea levels and the release of contaminants such as untreated sewage in coastal regions (Allard et al., 2020; Saintilan et al., 2020; Trevathan-Tackett et al., 2019). Therefore, understanding the microbial communities inhabiting the mangrove ecosystems and intricate interactions between these microbial communities and mangroves is essential for conservation and restoration projects. Furthermore, exploring microbial diversity is crucial for understanding their role in maintaining mangroves ecosystem and the biotechnological potential of mangroves microbiota (Imchen et al., 2018; Liao et al., 2020; Sivaramakrishnan et al., 2006; Thatoi et al., 2013).

Coastal areas of Umluj and Al-Wajh, within the Tabuk region of Saudi Arabia are located in the northwestern part of the country along the Red Sea and they are under gray mangrove (Avicennia marina) vegetation (Al-Guwaiz et al., 2021). The mangrove community of Red Sea is dominated by A. marina with the limited presence of Rhizophora mucronata (Blanco-Sacristán et al., 2022). The mangrove communities of the Red Sea are often characterized by lower and dwarf vegetation (Al-Guwaiz et al., 2021). Dwarfing in A. marina in contrast to mangroves reported in other regions is due to limited supply of nutrients (freshwater input), higher salinity, increased level of contaminants and elevated temperature (Almahasheer, Duarte & Irigoien, 2016; Anton et al., 2020; Mandura, 1997). Therefore, reliance on microbial communities for nutrient acquisition becomes crucial in these environments, highlighting the importance of understanding the distinct microbial populations that develop in response to these specific environmental conditions.

Though several studies in recent years have reported on the soil microbiome of various mangrove ecosystems, including the coastal areas of the Red Sea, the microbiome composition of the coastal areas of Umluj and Al-Wajh along the northeastern Red Sea coastline covered with A. marina vegetation has been overlooked (Alzubaidy et al., 2016; Basak et al., 2016; Iturbe-Espinoza et al., 2022; Ma et al., 2020; Ullah et al., 2017; Zhang et al., 2019). In this study, we planned to decipher the taxonomic diversity of bacterial communities found in the bulk soil and rhizosphere of A. marina, native to the coastal areas of Umluj and Al-Wajh. Specifically, we sought to determine whether the type of vegetation plays a crucial role in recruiting the rhizospheric microbiome and to identify key differences in the bacterial community composition between bulk and rhizosphere soil.

Materials and Methods

Study and sampling sites

The bulk soil and rhizosphere samples were randomly collected from two different areas Umluj and Al-Wajh along the coast of Red sea, in the Tabuk region, Saudi Arabia (Fig. S1). These coastal areas are predominantly covered by gray mangrove (Avicennia marina) vegetation. The mangrove vegetation in these regions is dwarfed compared to the mangroves reported in other areas (Anton et al., 2020). The sampling of the bulk and mangrove rhizospheric soil was performed in in December 2022. Three rhizosphere soil replicates (A. marina) and three bulk soil replicates samples were collected. For each replicate of the rhizosphere sample, three to five roots were collected by carefully uprooting A. marina plants, removing the soil loosely attached to the roots by shaking the plants, and then scraping and collecting the soil adhering firmly to the root by a sterile spatula, gathering approximately 10 g of soil (2 g from each root) at a depth of 20 cm. Three bulk soil samples were obtained from each site, approximately 7 m away from each mangrove tree, using a small trowel to dig soil from the top layer (0–15 cm). The collected soil samples were immediately stored in sterile plastic bags, placed in iceboxes, and brought back to the laboratory immediately. The samples were stored at −80 °C until DNA extraction.

DNA extractions

The total DNA was extracted from the rhizospheric and bulk soil samples. DNeasy® PowerSoil® Pro Kit (Qiagen, San Diego, CA, USA) was used to extract the total DNA from soil samples (1.0 g), following the manufacturer’s protocol. The quality and quantity of extracted DNAs were determined using a NanoDrop ND-1000 spectrophotometer (Thermo Fisher Scientific, Waltham, MA, USA) and DNAs with A260/280 ratio of 1.8–2.0 were used for subsequent experiments. Integrity and size of DNA were checked by 1% (w/v) agarose gel electrophoresis. DNA samples were stored at −20 °C.

16S rRNA amplicon sequencing

The total soil DNA samples from two locations were used for microbial diversity analysis based on 16S rRNA amplicon sequencing. PCR reactions and 16S rRNA libraries were prepared using 16S rRNA fusion primers targeting 16S rRNA V3–V4 region. All PCR products were purified by Agencourt AMPure XP beads, dissolved in elution buffer following the manufacturer’s protocol and eventually labelled to finish library construction. Library size and concentration are detected by Agilent 2100 Bioanalyzer. Qualified libraries were sequenced (300 x 2, pair end sequencing) using the DNBSEQ-G400 platform, a platform that amplifies and sequence with less error rates involving amplification of DNA fragments into DNA nanoballs (DNBs) which are then sequenced (BGI genomics, Shenzhen, China).

Taxonomical and functional analyses

Bioinformatic analyses were performed using the Fastq files generated after the sequencing runs. Raw data were filtered to obtain the high-quality clean data using iTools Fqtools fqcheck (v.0.25), cutadapt (v.2.6) and readfq (v1.0) (1). Raw data were filtered to generate high-quality clean reads as follows: (1) Reads with average Phred quality scores lower than 20 over a 25 bp sliding window were truncated. Reads were removed if their lengths were less than 75% of their original lengths after truncation, (2) reads contaminated by adapter sequences were removed using a default parameter of 15 bases overlapped by reads and adapter with a maximum of three bases mismatch allowed, (3) reads containing ambiguous bases (N bases) were removed and (4) low complexity reads, defined as reads with 10 consecutive identical bases, were removed. Barcode sequences were removed from pooled libraries, and clean reads were then assigned to corresponding samples based on barcode sequence alignments with no mismatches allowed.

The clean reads that can overlap with each other were merged (minimum overlapping length: 15 bp; mismatching ratio of overlapped region ≤0.1; FLASH tool, Fast Length Adjustment of Short reads, v1.2.11) into tags. Clean tags were clustered into OTUs using the UPARSE algorithm. Operational taxonomic units (OTUs) were identified to analyze taxonomic units in phylogenetic and population genetics research: (1) Sequences were clustered into OTUs at a 97% similarity threshold using UPARSE (USEARCH v7.0.1090), (2) chimeric sequences were filtered out using UCHIME v4.2.40, mapped to the gold database (v20110519) for 16S rDNA. All tags were mapped to OTU representative sequences using USEARCH GLOBAL to construct an OTU abundance table. OTU representative sequences were taxonomically classified with the RDP classifier (v2.2) with a sequence identity threshold of 0.6 using Greengene (default): V201305; RDP: Release 11.5 2016-9-30. Species composition and abundance were determined by aligning sequences against taxonomic databases, allowing the calculation of the relative abundance of each taxon in the samples. These analyses provided insights into the microbial community structure. Alpha diversity metrics (such as Shannon and Simpson indices) were calculated to assess the species diversity within each sample. Beta diversity was analyzed using principal component analysis (PCA) based on Bray-Curtis dissimilarity to evaluate differences in microbial community composition between samples using the R (v3.1.1; R Core Team, 2017) ade package. Functional prediction was performed using PICRUSt2, which infers the functional potential of microbial communities based on their phylogenetic placement. Pathway abundances were predicted and differentially abundant pathways between groups were identified using statistical methods. Spearman’s rank correlation coefficients were calculated for predicted functional pathways with a relative abundance greater than 0.5% and were visualized through heat maps to identify important patterns and relationships among dominant functional pathways. Differential analysis was performed using the Wilcoxon rank-sum test to identify predicted functional pathways with significant differences in relative abundance between groups. p-values and false discovery rates (FDR) were calculated, with significance set at p < 0.05.

Results

16S rRNA based amplicon sequence analysis

Bulk and rhizospheric soils under A. marina plants were sampled from two different locations: Ras-Alshabaan-Umluj (U) and Almunibrah-Alwajh (W), along the coasts of the Red Sea in the Tabuk region, Kingdom of Saudi Arabia. Metagenomic DNAs were extracted from the bulk and rhizospheric soils and subjected to 16S rRNA amplicon sequencing. A total of 1,504,842 raw reads (300 x 2) obtained from 11 DNA samples, were analyzed using various bioinformatic tools. First of all, quality control trimming and filtering yielded 1,496,294 good quality reads, averaging 136,026 sequences per sample and the reads were paired-up which generated sequences of average 415–420 bp in length (Table 1).

Table 1 Statistics of sequenced samples.

Site	Samples	Replicates	Raw reads	Paired reads	Average length	OTU number	
Umlij	Bulk soil (sample A)	CU1	137,078	68,210	415	877	
CU2	137,040	68,130	420	712	
CU3	136,624	67,966	416	582	
Al-Wajh	Bulk soil (sample B)	CW1	136,368	67,871	417	675	
CW2	136,070	67,636	419	729	
Umlij	Rhizosphere (sample C)	UA	136,604	67,845	419	836	
UB	137,376	68,241	420	2,630	
UC	137,090	68,115	417	921	
Al-Wajh	Rhizosphere (sample D)	WA	136,882	68,016	417	3,564	
WB	137,524	68,365	418	3,272	
WC	136,186	67,752	418	3,884	

The paired-up sequence reads exhibiting a similarity of ≥97% were clustered together into an OTU to quantify the abundance of bacteria at every taxonomic level (Phylum to species) in each sample. A total of 6,876 OTUs were obtained from all samples, of which only OTUs with ≥1% abundance in at least one of the samples are included in detailed analyses (Table 1).

Bacterial diversity in the bulk and rhizosphere soils from Umluj and Al-Wajh

OTUs based analysis was performed to identify the number of distinct and shared OTUs among different sites. The data showed that 678 OTUs were common to bulk and rhizosphere soils from Umluj, while 795 OTUs were common to bulk and rhizosphere soils from the Al-Wajh location (Fig. 1). These data suggest that the Al-Wajh location had more similar community structure between bulk soil and rhizosphere samples as compared to that of Umluj. While comparing the rhizosphere of both locations, the data showed 1,857 OTUs were common to the rhizosphere from Umluj and Al-Wajh. A. marina rhizosphere harbours a higher number of unique OTUs at Al-Wajh (3,011) as compared to the unique OTUs found in the rhizosphere of A. marina at Umluj (1,324). Overall, the number of OTUs found at both locations were higher in the rhizosphere samples as compared to the bulk soils of the respective locations.

Figure 1 The number of shared and unique operational taxonomical units (OTUs) observed in the samples from Umluj and Al-Wajh.

Bulk soil from Umluj (A), rhizosphere from Umluj (C), Bulk soil from Al-Wajh (B), rhizosphere from Al-Wajh (D) were analyzed based on the 16S rRNA amplicon sequence data as describes in material and methods.

These variations observed in bacterial communities in terms of the number of OTUs at different locations (Umluj and Al-Wajh) and habitats (bulk soil and rhizosphere) were verified by different diversity indices. Alpha diversity measured by Observed Species index, Chao index, Ace index, Shannon index and Simpson index revealed that diversity was significantly (p ≤ 0.05) different among the habitats (bulk soil and rhizosphere). The rhizosphere samples of both sites had overall higher bacterial diversity, richness and evenness compared to bulk soil samples (Fig. 2). All diversity indices showed significant higher diversity (p ≤ 0.05) for rhizosphere of Al-Wajh (Fig. 2B) as compared to the bulk soil of the same site. Similarly, in case of Umluj, most of the indices showed higher diversity in rhizosphere as compared to bulk soil, but in this case most diversity indices varied non-significantly (p > 0.05) except ace index (p < 0.05) (Fig. 2A). Comparing the diversity indices for the rhizosphere samples from the two sites revealed that the diversity was much higher in the rhizosphere samples from Al-Wajh as compared to those from Umluj (Fig. 2).

Figure 2 Alpha diversity indices measured for the bulk soil and rhizosphere samples from Umluj and Al-Wajh.

The diversity indices were measured from unique operational taxonomical units (OTUs) based on the 16S rRNA amplicon sequence data. The dot plots for each of the data sets Umluj bulk soil, Al-Wajh bulk soil, Umluj rhizosphere, and Al-Wajh rhizosphere are represented here. Statistical analyses result (pairwise students ttest) in terms of non-significant (ns, p < 0.05) or significant differences (*p < 0.05) are also presented. The rarefaction curves computed for alpha diversity index of the observed OTUs of individual samples are presented in Fig. S2.

The similarity and/or dissimilarity of microbial communities at rhizosphere and bulk soil samples from both sites was also measured. Our data showed that samples from bulk soils and rhizospheres from both sites occupied distinct positions (Figs. 3A and 3B). The microbial community in the bulk soil samples from Umluj was distinct from the microbial community in the rhizosphere samples of the same site (Fig. 3A). This difference was even more pronounced in the samples from the Al-Wajh site (Fig. 3B). Similarly, comparing the microbial diversities in the rhizosphere samples of A. marina from the two sites revealed the dissimilarities in their diversity as the distant placement of microbial community structure was observed in rhizosphere samples from Umluj and Al-Wajh (Fig. 3C).

Figure 3 PCA plots showing beta diversity in bulk soil and rhizosphere samples from Umluj and Al-Wajh.

Diversity was compared between bulk and rhizosphere samples from Umluj (A), bulk and rhizosphere samples from Al-Wajh (B) and rhizosphere samples from Umluj and Al-Wajh (C). Operational taxonomical units (OTUs) based on the 16S rRNA amplicon sequence data were used for these analyses. R (v3.1.1) ade package was used for PCA based on Bray-Curtis dissimilarity (with PERMANOVA analysis) to evaluate differences in microbial community composition and for visualisation.

Distribution of bacterial taxa in the bulk and rhizosphere soils from Umluj and Al-Wajh

Detailed sequence analysis involving the identification of OTUs into different taxa and the distribution of these taxa in different samples, revealed that the most abundant phyla in both soil and rhizosphere samples from Umluj were Actinobacteria, Proteobacteria, Firmicutes, Bacteriodetes and Acidobacteria, and the most abundant phyla in both soil and rhizosphere samples from Al-Wajh were Firmicutes, Acidobacteria and Chloroflexi (Fig. 4). Interestingly, relative abundance of Actinobacteria is lower in the rhizosphere samples as compared to that in their respective bulk soil samples at both Umluj and Al-Wajh sites, and the relative abundance of Firmicutes is lower in the rhizosphere samples from Al-Wajh as compared to that in the respective bulk soil sample (Fig. 4). The relative abundance of Proteobacteria, Bacteriodetes, Acidobacteria, Fusobacteria and Chloroflexi are higher in the rhizosphere samples from Umluj as compared to that in the respective bulk soil samples (Fig. 4), while the relative abundance of Proteobacteria, Acidobacteria, Bacteriodetes, Nitrospirae and Chloroflexi are higher in the rhizosphere samples from Al-Wajh as compared to that in the respective bulk soil samples (Fig. 4).

Figure 4 Bacterial phyla observed in the bulk and rhizosphere soil samples from Umluj and Al-Wajh.

Operational taxonomical units (OTUs) abundance table based on the 16S rRNA amplicon sequence data were used for taxonomic affiliation analyses at phyla level as described in material and methods. Relative abundance of various phyla distributed among the bulk and rhizosphere samples from Umluj and Al-Wajh is presented here.

Analyses at the class level revealed that among the top 20 classes present at abundances exceeding 1% in one or more samples, Actinobacteria, Alphaproteobacteria, Betaproteobacteria, Gammaproteobacteria, Deltaproteobacteria, Sphingobacteriia, Clostridia, Anaerolineae, Bacilli, and Acidobacteria contributed significantly to the overall bacterial diversity regardless of the sampling site (Fig. 5). Interestingly, the relative abundance of Actinobacteria, Alphaproteobacteria, and Betaproteobacteria was lower in the rhizosphere samples from Umluj compared to the respective bulk soil sample, while the relative abundance of Gammaproteobacteria, Deltaproteobacteria, Sphingobacteriia, Acidobacteria, Bacteroidia, and Nitrospira was higher in the rhizosphere samples from Umluj compared to the bulk soil samples from the same site (Fig. 5). However, Bacilli were dominant in both bulk soil and rhizosphere samples from Umluj. Similarly, the relative abundance of Actinobacteria, Betaproteobacteria, Gammaproteobacteria, Bacilli, and Clostridia was higher in the bulk soil samples from Al-Wajh, while the relative abundance of Alphaproteobacteria, Deltaproteobacteria, Anaerolineae, and Nitrospira was higher in the rhizosphere samples from Al-Wajh compared to the bulk soil samples from the same site (Fig. 5). Comparing the rhizosphere samples from the same type of plants growing at different sites revealed that several bacterial classes had almost similar relative abundance at both sites. However, the relative abundance of Actinobacteria, Bacilli, Sphingobacteriia, Acidobacteria Gp26, and Bacteroidia was higher in the rhizosphere samples from Umluj compared to those in the rhizosphere samples from Al-Wajh. Conversely, the relative abundance of Deltaproteobacteria, Anaerolineae, and Nitrospira was higher in the rhizosphere samples from Al-Wajh compared to those in the rhizosphere samples from Umluj (Fig. 5).

Figure 5 Bacterial classes observed in the bulk and rhizosphere soil samples from Umluj and Al-Wajh.

Operational taxonomical units (OTUs) abundance table based on the 16S rRNA amplicon sequence data were used for taxonomic affiliation analyses at class level as described in material and methods. Relative abundance of bacterial classes distributed among the bulk and rhizosphere samples from Umluj and Al-Wajh is presented here.

To enhance our understanding of the most abundant taxa of bacterial communities found in the bulk and the rhizospheric soils of A. marina from both sites, diversity analysis revealed that composition of bacterial communities varied significantly depending on the site and habitat. The bulk soil samples from Umluj were dominated by Arthrobacter, Blastococcus, and Pseudomonas while the most abundant genera inhabiting the rhizospheric soil of the same site were Propionibacterium, Corynebacterium, Staphylococcus and Acidobacteria Gp10 and GP26 (Fig. 6). Of all the dominant genera, only Pseudomonas and Pelagibius occupied similar relative abundance in both habitats at the Umluj site. Similarly, the bulk soil samples from Al-Wajh were dominated by Pseudomonas, Arthrobacter, Anaerococcus, Strenotrophomonas and Propionibacterium while Geminicoccus and Thermodesulfovibrio were abundant in the rhizospheric samples. Geminicoccus, was the only genus, inhabiting both the rhizosphere and the bulk soil samples of Al-Wajh. In general, the relative abundance of Propionibacterium was higher in the rhizosphere and bulk soil samples of Umluj and Al-Wajh respectively. Interestingly, the rhizospheric bacterial community differed at both Umluj and Al-Wajh sites, regardless of the type of vegetation. The relative abundance of Propionibacterium, Staphylococcus, Acidobacteria Gp10, Gp26, Prevotella, Streptococcus and Corynebacterium was higher in the rhizosphere samples of Umluj compared to the rhizosphere samples of Al-Wajh. In contrast, Geminicoccus and Thermodesulfovibrio were the most abundant taxa present in the rhizosphere samples from Al-Wajh.

Figure 6 Relative abundance and distribution of most abundant bacterial species observed in the bulk and rhizosphere soil samples from Umluj and Al-Wajh.

Relative abundance of species distributed among the bulk and rhizosphere samples from Umluj (A), bulk and rhizosphere samples from Al-Wajh (B) and rhizosphere samples from Umluj and Al-Wajh (C) is presented here.

Predicted functional diversity in the bulk and rhizosphere soils from Umluj and Al-Wajh

The functional profiles of bacterial communities inhabiting the bulk soil and rhizosphere of A. marina from both sites were predicted using Picrust2 (Phylogenetic Investigation of Communities by Reconstruction of Unobserved States). Based on the prediction results, bacterial taxa and their functions can be linked to obtain general distribution profiles of community function. Picrust2 predicts the function (including the prediction of KEGG, COG, and MetaCyc metabolic pathways) abundance of microbial communities based on 16S rRNA sequencing profiles (Douglas et al., 2020). Among functional categories predicted using MetaCyc database in the samples from both sites, amino acid biosynthesis and degradation, nucleoside and nucleotide biosynthesis and degradation, cofactor/prosthetic-group/electron-carrier/vitamin biosynthesis, carbohydrate biosynthesis and degradation, fatty acid and lipid biosynthesis, C1 compound utilization and assimilation, cell structure biosynthesis, TCA cycle, fermentation, and secondary metabolite biosynthesis, were abundant in both bulk soil and rhizosphere samples from both sites (Fig. 7).

Figure 7 Predicted functional diversity observed in the bulk and rhizosphere soil samples from Umluj and Al-Wajh.

MetaCyc pathway profiles, encompassing primary and secondary metabolism along with associated metabolites, reactions, enzymes, and genes, were generated using PICRUST2. These pathways were categorized based on their biological functions and the types of metabolites they either produce or consume. The MetaCyc level two metabolic pathway abundance heatmap presented here was created using the ‘pheatmap’ package in R (v3.4.10). The heatmap employs the ‘euclidean’ distance algorithm and ‘complete’ clustering method.

The relative abundance of microbial genes varied depending on the habitat. Differential functional abundance analyses revealed that only a few functions were enriched with a slightly significant difference (slightly significant as the p < 0.1) in the rhizosphere as compared to the bulk soil samples (Fig. 8). The functions that were more abundant in the bulk soils included chlorinated compound degradation, nucleoside and nucleotide degradation, carboxylate degradation, Entner−Duodoroff pathways, aromatic compound degradation, cofactor/prosthetic-group/electron-carrier/vitamin biosynthesis, alcohol degradation, cell structure biosynthesis, and the TCA cycle. Interestingly, the functions related to C1 compound utilization and assimilation were more enriched in the rhizosphere samples as compared to the bulk soil.

Figure 8 Differential functional abundance analysed in the rhizosphere as compared to the bulk soil samples.

The Wilcoxon test was applied to predicted functions to identify differential functions among all groups using R (v3.4.1). Horizontal bars representing differential relative abundance are shown on the left, log2 values of the ratio of average relative abundance between the two groups are displayed in the middle, and p-values and FDR values calculated from the Wilcoxon test are shown on the right.

Discussion

Exploring the microbial community composition associated with the roots of A. marina in coastal areas is crucial for understanding their ecological role in maintaining mangrove ecosystems and their potential to sustain plant growth. While several studies have reported the soil microbiome of different mangrove ecosystems in recent years, the coastal areas of the Red Sea, especially the northeastern coastline of Red Sea with A. marina vegetation have largely been overlooked in this regard (Basak et al., 2016; Iturbe-Espinoza et al., 2022; Ma et al., 2020; Zhang et al., 2019). We explored the bacterial community at the Umluj and Al-Wajh sites in the Tabuk region using 16S rRNA amplicon sequencing. The bacterial diversity of bulk soil and rhizosphere of A. marina from these sites was compared to identify the key differences in the community composition and to understand the role of A. marina in the recruitment of microbial communities in the rhizosphere.

Microbial diversity analysis of the bulk and rhizospheric soil samples from Umluj revealed an abundance of the phyla Actinobacteria, Proteobacteria, Firmicutes, Bacteriodetes and Acidobacteria. In contrast, Firmicutes, Acidobacteria and Chloroflexi were the most abundant phyla found in the bulk soil and rhizospheric soil samples from Al-Wajh, suggesting the impact of location on the overall composition of bacterial communities. The impact of geographical location on the bacterial composition can be associated with the environmental factors affecting the soil composition. Various physiochemical factors of mangrove soil, such as pH, electrical conductivity, salinity and nutrient contents support the growth of a wide range of bacterial communities based on the ecological environment (Hossain, Aziz & Saha, 2012; Hu et al., 2022; Nimnoi & Pongsilp, 2022). Other environmental factors, such as urban development, pollution and seasonal changes, are also reported to play a significant role in shaping the bacterial composition in mangrove soil (Quintero, Castillo & Mejía, 2022).

Marine ecosystems and coastal waters might also impact the overall bacterial community composition of coastal areas. Actinobacteria one of the most dominant phyla and classes at the explored sites, was found in more abundance in bulk soil as compared the rhizosphere soils at both sites. This suggests that it might not be A. marina plant that contributes to the presence of Actinobacteria; instead, marine ecosystems and coastal waters may be the source of these bacteria. Actinobacteria are not only one of the most abundant bacterial taxa in marine microbial communities but also highly diverse in various marine ecosystems (Kavitha & Savithri, 2017; Ribeiro et al., 2020; Sottorff, Wiese & Imhoff, 2019; Yu et al., 2015). Among Proteobacteria, the other most abundant phyla at the explored sites, the classes Alphaproteobacteria, Gammaproteobacteria, and Deltaproteobacteria were predominant. Alphaproteobacteria and Gammaproteobacteria were distributed without any clear preference for bulk or rhizospheric soils under A. marina vegetation. For example, Gammaproteobacteria were the most abundant class present in the rhizosphere samples from Umluj, whereas the same bacterial class was abundant in the bulk soil samples from Al-Wajh, where Alphaproteobacteria predominated compared to the rhizosphere samples. Previously, both Alphaproteobacteria and Gammaproteobacteria have been reported as predominant taxa in samples from Chinese and Brazilian mangrove ecosystems, with mean annual precipitation, total organic carbon, and total nitrogen identified as key factors influencing their abundance (Andreote et al., 2012; Liang et al., 2007; Zhang et al., 2019).

Deltaproteobacteria were abundant in the rhizosphere samples as compared to the respective bulk soils from both sites, suggesting that this class may be the part of core microbiome of mangrove ecosystems as suggested in previous studies (Zhang et al., 2019). The prevalence of Deltaproteobacteria in the mangrove ecosystems can be attributed to its metabolic flexibility, which provides a competitive advantage for surviving in fluctuating and harsh environments (Dombrowski, Teske & Baker, 2018; Rath et al., 2019; Trivedi et al., 2016). Additionally, higher levels of Deltaproteobacteria in the rhizosphere of A. marina might be associated with anaerobic conditions within mangrove sediments, supporting anaerobic microbial communities including sulfur-reducing bacteria (Alzubaidy et al., 2016; Nathan, Vijayan & Ammini, 2020; Thatoi et al., 2013; Zhuang et al., 2020).

In addition to Deltaproteobacteria, the predominance of classes such as Sphingobacteriia, Anaerolineae, Acidobacteria, Bacteriodetes, Chloroflexi and Nitrospirae in the rhizosphere samples from both sites as compared to the respective bulk soil samples is consistent with the rhizosphere effect (Ismail et al., 2017; Ullah et al., 2017). The abundance of Bacteroidetes in the rhizospheric soil compared to bulk soil samples is in agreement with previously reported studies (Gomes et al., 2010). They have been reported to be abundant in inter-tidal regions, particularly in hydrocarbon-contaminated regions, known for their potential to mineralize high-molecular-weight organic matter (Kim & Kwon, 2010; Kirchman, 2002). The relative abundances of Chloroflexi and Nitrospirae was higher in the rhizospheric soil samples of Umluj and Al-Wajh, suggesting the potential of organic matter decomposition and nitrogen metabolism in that habitat (Pinto et al., 2016; Wang et al., 2012; Yamada et al., 2005). Our findings align with another study that used amplicon and metagenome sequencing to reveal distinct microbial communities in different root compartments (nonrhizosphere, rhizosphere, episphere, endosphere) of mangroves (Zhuang et al., 2020). Zhuang et al. (2020) found unique distribution patterns for bacterial and fungal communities due to niche differentiation and root exudation, highlighting the importance of soil-root interfaces in shaping microbial diversity and function in mangrove ecosystems. Root exudates, serving as a rich source of carbon for microbes, play a significant role in shaping the microbial communities around plant roots (Ma et al., 2022). Meanwhile communities inhabiting the rhizosphere assist plants in obtaining essential nutrients such as phosphorus, potassium and nitrogen (Hu et al., 2018; Vives-Peris et al., 2020).

In our study, the rhizospheric samples from Umluj were dominated by Propionibacterium, Corynebacterium, Staphylococcus and Acidobacteria (Gp10 and Gp26). The presence of Propionibacterium in rhizosphere of A. marina is surprising as its association with the roots of A. marina has not been reported until now. However, the abundance of Corynebacterium in the rhizospheric soil sample of Umluj revealed increased nitrogen cycling possibility in the rhizospheric soil in contrast to bulk soil samples. Two different strains of Corynebacterium; strain 63K and 12A being associated with roots of Avicennia germinans, have been reported to be involved in the process of denitrification (Flores-Mireles, Winans & Holguin, 2007). Another study reported the abundance of Corynebacterium sp. in mangrove sediments and its involvement in the decomposition of mangrove leaf litter (Yulma et al., 2020). Apart from their degradation potential, they are known for pathogenicity in Rhizophora mangle, a red mangrove (Ukoima, Wemedo & Ekpirikpo, 2009). Moreover, the relative abundance of Staphylococcus, a nitrogen-fixing bacteria, in the rhizospheric soil of Umluj is consistent with a previous study that reported the isolations of Staphylococcus from the rhizosphere of mangrove trees (Holguin, Guzman & Bashan, 1992). They have been found to actively participate in the process of decomposition (Ogbonna, 2011). Apart from their role in decomposition, Staphylococcus xylosus, isolated from petroleum contaminated soil, have been reported for their potential to produce biosurfactant. The higher abundance of Acidobacteria (Gp10 and Gp26) in the rhizospheric soil samples of Umluj is also in concordance with a recent report in which Acidobacteria were found abundant in the Pneumatophore-associated soil microbial community of A. marina mangrove (Sanka Loganathachetti et al., 2016).

The rhizosphere samples from Al-Wajh as compared to the bulk soil samples from the same site were dominated by Thermodesulfovibrio, a genus of sulfur-reducing bacteria, which revealed the significant role of geochemical properties in shaping the microbial communities around the roots. Thermodesulfovibrio, a thermophilic sulfate-reducer, plays a significant role the biogeochemical processes, particularly reducing sulfates into sulfides (Haouari et al., 2008). Thermodesulfovibrio yellowstoni, an anaerobic sulfate reducer, was the one of most abundant species in mangrove sediments of Yunxiao Mangrove National Nature Reserve China (Lin et al., 2019). Sulfur-reducing bacteria such as Desulfococcus and Desulfosarcina, along with their associated functional genes, have recently been reported in abundance in the rhizosphere of Kandelia obovata, a native mangrove plant of Southern China (Zhuang et al., 2020), probably due to the presence of oxygen and redox potential gradients facilitating sulfate reduction in the rhizosphere. Surprisingly, the abundance of Geminicoccus in the rhizospheric soil samples of Al-Wajh is contradictory as there is no such literature available about the association of Geminicoccus with A. marina roots. However, a recent study has reported the potential of Geminicoccus roseus to produce biosurfactants in oil-contaminated soils (Saisa-Ard, Saimmai & Maneerat, 2014). Two different species of Geminicoccus, G. flavidas and G. harenae, inhabiting the desert soil crust are known for their ability to promote plant growth via IAA production (Jiang et al., 2022).

There were certain bacterial genera which showed higher relative abundance in bulk soil samples as compared to their relative abundance in rhizosphere samples from Umluj. These included Pseudomonas and Arthrobacter, suggesting the potential of the microbiota in hydrocarbon degradation and degradation of other toxic substances respectively in these soils (Aboelkheir et al., 2019; Baig et al., 2022; Desai, Pathak & Madamwar, 2010; Zhao et al., 2018). The abundance of Pseudomonas in soils from coastal areas is in consistent with a previous study that reported the isolation of two different strains of Pseudomonas aeruginosa in samples from Great Nicobar (Kothamasi et al., 2006). Various species of Pseudomonas have been known for their ability to degrade hydrocarbon and polythene bags (Brito et al., 2006; Kathiresan, 2003). Additionally, a study has reported the capacity of Arthrobacter sp. to degrade Acetaminophen, an anti-inflammatory drug in mangroves sediments. Apart from their role in bioremediation, Arthrobacter has been also reported to play an important role in the process of denitrification in coastal ecosystems (Flores-Mireles, Winans & Holguin, 2007). The relative abundance of Blastococcus in the bulk soil of Umluj is also in agreement with the study in which it has been isolated from mangrove soil in Leizhou Peninsula (Lu et al., 2021). Another specie of Blastococcus, Blastococcus litoris have been isolated from sea-tidal flat sediments (Lee et al., 2018). Blastococcus, an actinobacteria is generally a part of soil microbiome having potential resilience to the presence of heavy metals (Chouaia et al., 2012; Godoy-Lozano et al., 2018; Pereira, Vicentini & Ottoboni, 2015; Wang et al., 2023).

Similarly, certain bacterial genera showed higher relative abundance in bulk soil samples as compared to their relative abundance in rhizosphere samples from Al-Wajh. In addition to Arthrobacter and Pseudomonas, the relative abundance of Stenotrophomonas, and Anaerococcus in the bulk soil samples of Al-Wajh suggested a potential bioremediation activity. Interestingly, a study reported the polycyclic aromatic hydrocarbon biodegradation potential of genus Stenotrophomonas isolated from coastal sediments (Aziz et al., 2018). In addition, another species of Stenotrophomonas, Stenotrophomonas rhizophila, isolated from rhizosphere of Kandelia cande, has been shown to slow down the process of eutrophication (algicidal effect) (Zhang, Wang & Zhou, 2021). The prevalence of bio-degraders in coastal ecosystems is due to a large number of contaminants being released in these areas, including untreated sewage, accidental oil spillage and industrial effluents (Aziz et al., 2018). The abunadance of Anaerococcus and Propionibacterium in bulk soil samples was surprising as these are commonly human associated Gram-positive bacteria (Morand et al., 2021; Nazipi et al., 2017). These bacteria might have been introduced into mangrove ecosystems through various anthropogenic activities (Murphy & Frick, 2013). They are usually categorized into two groups as, cutaneous and classical Propionibacterium suggesting a complex interplay between human activities and mangrove ecosystems. Human pathogenic microbes may find their way to such ecosystems through urban runoff and wastewater discharges (Marques et al., 2023).

Though the difference in the microbial communities at taxonomic level was observed, almost similar patterns of predicted functional genes were noted for both sites based on the Metacyc functional profiling. However minor differences in the abundance of microbial genes were observed on the basis of habitat (Fig. 8). The relative abundance of genes for C1 compound utilization and assimilation in the rhizosphere coincides with the presence of Methylobacterium and Methyloceanibacter in samples from Umluj and Al-Wajh respectively. Anaerobic conditions in the mangrove sediments make them hotspot for C1 metabolisms as methanogenic activities will result in methane emissions (Dar et al., 2008; Lyimo, Pol & den Camp, 2002). Moreover, methanol is also released from roots and leaf litter (Folkers et al., 2008; Oikawa et al., 2011). The presence of methane and methanol might promote the recruitment of methylotrophs and methanotrophs in oxic layers of mangrove sediments (do Carmo Linhares et al., 2021; Shiau et al., 2018; Zhuang et al., 2020). The absence of oxygen and abundance of organic matter, creates an ideal environment for various groups of microorganisms including methanogens, methanotrophs, acetogens and sulfate reducers which are capable of catabolizing toxic carbon compounds (Alzubaidy et al., 2016; Dar et al., 2008; Lyimo et al., 2009). Similarly, the genes involved in the degradation of chlorinated compounds were abundant in bulk soil as mangrove soils are the major natural sinks for anthropogenic and industrial pollutants with enhanced decomposition potential (Bayen, 2012). These result which are in line with the results of alpha diversity analysis, showing Pseudomonas as a predominant genus in the bulk soil samples of both sites as it has been reported to degrade several chlorinated compounds under anaerobic conditions (Nikel, Pérez-Pantoja & de Lorenzo, 2013). While functional potential prediction analyses provide valuable insights, they have limitations, including the percentage of OTU/reads predicted, which can affect the accuracy of the functional annotations. Incorporating metagenomic approaches would provide a more comprehensive understanding of the functional roles of bacterial communities.

In addition to niche differentiation and root exudation underscoring the importance of soil-root interfaces in shaping microbial diversity in mangrove ecosystems, seasonal dynamics and ecological parameters like urbanization and chemical pollution greatly influence changes in bacterial diversity and composition in these ecosystems. Umluj and Al-Wajh, both part of the Tabuk region, have unique geographical and environmental features. Umluj is known for its pristine beaches and coral reefs, contributing to its biodiversity and Al-Wajh, on the other hand, has a different set of environmental conditions due to variations in sediment composition and coastal processes (Al-Hashim et al., 2021) including higher anthropogenic influences like fishing, agriculture and pollution. The unique OTUs and microbial community compositions observed in each site might have been influenced by these factors. Similarly, a reduced microbial diversity have been reported in anthropogenically affected mangrove when compared to that of pristine mangrove suggesting an important role of environmental regulators in shaping the composition of microbial communities in mangrove ecosystems (Haldar & Nazareth, 2018). Another factor that could have contributed to the bacterial distribution and functionality in mangrove soils is bioturbation. Burrowing crabs and other crustaceans, which are strongly associated with mangrove trees, counteract the reduced redox potential in mangrove sediments through their burrowing activities, ultimately altering microbial composition and function significantly in mangrove ecosystems (Ferreira et al., 2007; Gillis et al., 2019). Increased bioturbation enhances the presence of bacteria that are involved in carbon, nitrogen and phosphate cycling in mangrove ecosystem of Red Sea coast thus ultimately improving the plant growth particularly under extreme summer conditions (Fusi et al., 2022). Additionally, seasonal variation also plays an important role in altering the bacterial diversity of mangrove ecosystem as depicted by another study. During the monsoon the mangrove bacterial community was dominated by Actinobacteria, which shifted to Gammaproteobacteria in summer (Behera et al., 2019; Palit et al., 2022). Overall, the changes in the carbon, nitrogen and sulfur contents along with changes in pH and salinity could have been attributed to variations in the bacterial community structure in two mangrove ecosystems.

Conclusion

Altogether our study provides a new insight about the differences in the composition of microbial communities in the rhizospheres of gray mangrove vegetations from Umluj and Al-Wajh, within the Tabuk region of Saudi Arabia. We have found that microbial communities inhabiting the rhizosphere samples were significantly different from those inhabiting the bulk soil suggesting the possible role of A. marina in defining the soil microbial community. In addition, the structural composition of bacterial community also depends on several other factors such as geographical location and ecological parameters, as revealed by differences in bacterial communities between soils from Umluj and Al-Wajh despite similar vegetation. Mangrove microbiomes play a crucial role in nutrient cycling processes, which are frequently disrupted by anthropogenic activities. Further research into functional gene analysis and whole metagenome analyses of microbial communities in these mangrove ecosystems is essential for understanding the role of microbes in maintaining mangrove ecosystems and exploring the biotechnological potential of mangrove microbiota. Additionally, studying the impact of chemical pollution on identified microbial communities could provide valuable insights into the adaptability of mangrove ecosystems and contribute to the development of effective conservation strategies.

Supplemental Information

Supplemental Information 1 Ras Alshabaan-Umluj (Umluj), and Almunibrah-Al-Wajh (Al-Wajh) Tabuk region, Saudi Arabia.

Figure S1 photo credit: Hanaa Ghabban. Figure S2 photo credit: Doha A. Albalawi.

Additional Information and Declarations

Competing Interests

Author Contributions

Data Availability

The authors declare that they have no competing interests.

Hanaa Ghabban conceived and designed the experiments, performed the experiments, analyzed the data, prepared figures and/or tables, authored or reviewed drafts of the article, and approved the final draft.

Doha A. Albalawi conceived and designed the experiments, performed the experiments, analyzed the data, prepared figures and/or tables, authored or reviewed drafts of the article, and approved the final draft.

Amenah S. Al-otaibi conceived and designed the experiments, performed the experiments, analyzed the data, prepared figures and/or tables, authored or reviewed drafts of the article, and approved the final draft.

Dikhnah Alshehri conceived and designed the experiments, analyzed the data, prepared figures and/or tables, and approved the final draft.

Asma Massad Alenzi conceived and designed the experiments, analyzed the data, prepared figures and/or tables, and approved the final draft.

Marfat Alatawy conceived and designed the experiments, analyzed the data, prepared figures and/or tables, and approved the final draft.

Hanan Ali Alatawi conceived and designed the experiments, analyzed the data, prepared figures and/or tables, and approved the final draft.

Dalia Kamal Alnagar analyzed the data, prepared figures and/or tables, and approved the final draft.

Ahmad Bahieldin conceived and designed the experiments, authored or reviewed drafts of the article, and approved the final draft.

The following information was supplied regarding data availability:

The sequences are available at NCBI: PRJNA1077244.

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
