# Peer review of "Investigating the bacterial community of gray mangroves (Avicennia marina) in coastal areas of Tabuk region"

_PeerJ, doi:10.7717/peerj.18282_

## Round 0.1 · original submission · Major Revisions

As two reviewers comment, the manuscript has validity if strongly corrected. Please follow all the suggestions, including reviewer 1 concerns.

Reviewer 1 ·

Basic reporting

The authors performed a comparative analysis using 16s RNA between the microbial community of two mangroves in Saudi Arabia considering two types of communities, the rhizosphere and the soil.

Despite interesting results such as differences in the community structure of soil and rhizosphere, the analyses presented in the article are very superficial and are not explained in any way in the methods section. Additionally, the introduction does not delve deeper into the possible differences between the rhizosphere and soil microbiota, and the pathway analysis is inconclusive. Therefore, I consider that the paper needs to work more on its results and strengthen its conclusions, primarily by clarifying its methods. Hence, I recommend rejection.

Experimental design

The question posed in the introduction is confusing. It is not clear how this work contributes; its main contribution is that the rhizosphere is different from the soil, but this is lost due to lack of clarity.

The authors do not describe their methods clearly; it is not possible to know how the results they present were generated.

Validity of the findings

Due to the lack of clarity in both the research question and methodology, the results lack reproducibility, and the conclusions lack clear novelty.

Additional comments

The authors should clarify their methodology, especially regarding statistics, to ensure the reproducibility of their results. The article would benefit from a clearer research question and a more direct, less vague introduction. I suggest that the functional analysis focus on the primary biogeochemical cycles and pathways associated with anthropogenic pollution and salinity.

Reviewer 2 ·

Basic reporting

This study used the 16S rRNA amplicon sequencing to investigate the bacterial community diversity, community composition and potential function across the bulk and rhizospheric soils associated with Avicennia marina plants. Although the experimental design and amplicon sequencing technology lack innovation, the information on the bacterial composition in specific mangrove species (Avicennia marina) has still enhanced our understanding of mangrove symbiotic microorganisms. In addition, I think there are multiple instances of incorrect usage of professional terminology and errors in English writing in the article. For example, line 151 Metagenomics DNA sequencing The way it's presented is incorrect, as there's no mention of metagenomic sequencing throughout the entire article; it should be changed to "16S rRNA amplicon sequencing." The extensive use of terms like "metagenomics" or "metagenomic" throughout the text could be misleading, as "metagenomics sequencing" and "amplicon sequencing" are two entirely different techniques. In the discussion section, it seems that the authors may have overlooked key recent literature on root-associated microbial communities (Zhuang, Wei, et al. "Diversity, function and assembly of mangrove root-associated microbial communities at a continuous fine-scale." npj Biofilms and Microbiomes 6.1 (2020): 52). The authors should compare the differences in bacterial community diversity with the root-associated bacterial community of Kandelia obovata mangroves.
Therefore, I suggest that the article needs substantial revision and finding a fluent English speaker for language editing.

Experimental design

The described experiment in the manuscript is reasonable, and the overall logic is well defined. However, the authors did not describe the specific analysis steps and visualized tools used for the PCA in fig3, significant difference analysis of boxplot in fig2, and heatmap in fig7 and barplot in fig8 presented in the Results section. This information should be supplemented in the Materials and Methods section.

Validity of the findings

There have been numerous studies on the symbiotic microorganisms in mangrove roots, with researchers investigating root microbial communities worldwide using both culture-dependent and sequencing-dependent methods (including bacteria, archaea, fungi, etc.). However, research on whether different species of mangroves harbor distinct microbial communities is still ongoing. This is important because different types of mangroves may have varying ecological impacts on coastal wetland ecosystems globally. Evaluating the unique root microbiota and their functions in different types of mangroves can help researchers select and cultivate mangrove species that contribute to greenhouse gas reduction and mitigate coastal pollution. This study could provides a theoretical basis for further management and exploration of microbial resources in mangrove ecosystems.

Reviewer 3 ·

Basic reporting

General Review:
The manuscript presents an investigation into the bacterial community associated with Avicennia marina in the Red Sea coastal areas, shedding light on their structure, diversity and functionality predicted from taxonomic diversity. The manuscript effectively communicates the main findings, including insights into bacterial diversity patterns, dominant genera in rhizospheric soil samples, and functional potential prediction analyses. These findings contribute to our understanding of microbial community dynamics in mangrove arid, oligotrophic ecosystems.

Experimental design

See comments below

Validity of the findings

See comments below

Additional comments

General comments:
- Substitute “microbial” with “bacterial” in the title.
- It is recommended that the authors carefully review the English grammar and structure of the manuscript to ensure clarity and coherence throughout the text.
- If the main aim of this work is to explore the “functional” potential of the mangrove-associated bacterial community, the authors should apply metagenome sequencing that allows the analysis of the “genomic metabolic/functional potential” encoded by bacteria. The data presented here are limited to “taxonomic diversity” based on 16S rRNA gene amplicon sequencing. Please revise the aim according to the data/results presented.
- Please, clarify in the manuscript the hypothesis of the study. It would be beneficial to guide readers in understanding the research objectives and the direction of the investigation.
- Ensure that the references are up to date and accurately reflect the current literature relevant to the study’s topic.
- Why the recruitment of a different community associated with the rhizosphere should be a novelty? This has been observed in all terrestrial and marine plants so far. Did you evaluate how the recruitment occurred? Is there a “selection” process? Does this imply an enrichment of specific bacteria? More information regarding the recruitment process mediated by the plant and its consistency across sites is important to straighten the results reported here.
- Why the association with bacteria should be particularly important in mangroves such as those inhabiting the Red Sea coast? Recent papers highlighted the oligotrophic conditions of such sediments, including their nutrient limitation, for instance, indicating how these plants are much smaller than those of the same species growing in other countries; among others, 10.3389/fmars.2020.00597.
- Ensure that all necessary details are provided in the methods section. It appears that some crucial information may be missing (among others, the number of replicates collected, sequencing, rarefactions, and controls); including these details will improve the reproducibility and transparency of the study.
- Are three replicates (and in one case two, see Table 1) enough to describe the heterogeneity of mangrove ecosystems? Moreover, how, in the case of a test including samples with two replicates, the statistical analysis can be run properly?
- The authors mention the influence of geographical location on bacterial composition but could provide additional details on the factors contributing to this variation. Discussing potential environmental drivers or site-specific conditions could enrich the interpretation of results. Did you have any possible clue related to climatic conditions/physico-chemical conditions of sediments, and biome diversity? See, for example, recent papers on the effect of crab bioturbation on mangrove development, nutrient cycle and microbial diversity.
- While the manuscript presents data on bacterial diversity and predicted functional potential, providing additional context to interpret these results would strengthen the discussion section. Discussing potential ecological implications or linking findings to existing literature could enrich the interpretation of results.
- The manuscript briefly mentions the influence of geographical location on bacterial composition. Expanding on this aspect by discussing potential environmental drivers or site-specific conditions contributing to microbial variation would enhance the depth of analysis. For instance, while the authors mention the total number of OTUs and those common to all locations, it would be beneficial to provide more context on the significance of the unique OTUs observed in each site. This could include discussing potential ecological implications or specific taxa associated with these unique OTUs.
- The authors rely on the functional potential prediction analyses, but they should elaborate further on the limitation of this approach (e.g., which is the percentage of OTU/reads predicted). Providing insights into the functional roles of bacterial communities based on a metagenomic approach would enhance the significance of the study.
- Including suggestions for future research directions or implications of the study's findings would add value to the conclusion section. Identifying areas for further exploration or addressing any limitations of the current study could guide future investigations in this field.

Specific comments:
- Please specify “predicted” functional diversity since data were predicted from taxonomic diversity and not obtained from metagenomes. Modify it throughout the entire manuscript.
- Lines 107-109. Several studies analyse the microbiome of sediment in Red Sea mangroves. Please rephrase and update the literature. Among them, works based on amplicon sequencing were also done.
- Line 137 “Marina” is “marina”. Check along the manuscript.
- Line 137. How do you define the soil-sampled rhizosphere? Generally, the rhizosphere is the soil “attached” to the roots. If the root were not collected, it is better to define this soil as “root surrounding soil” or “soil affected by root” I would avoid using the term rhizosphere here.
- Line 140. What was the density of mangroves? Which was the other vegetation present (if any)? 7 m in which direction (sea or land)? Please describe each site before describing the sampling procedures and provide representative pictures in supplementary material.
- Line 146, invert quality and quantity.
- Line 151. Since PCR has been performed to amplify the 16S rRNA gene before sequencing, for clarity, I suggest using the title (and in the text) “Bacterial 16S rRNA gene amplicon sequencing” instead of “Metagenomics DNA sequencing”.
- Line 158. How was the library sequenced? Please add details, including reads obtained and accession numbers of data submitted to public repositories, such as NCBI/SAR.
- Did you use and sequence any controls? Such as PCR negative control and blank of extraction, including all reagents of DNA extraction kits (but no samples).
- Provide rarefaction curves after the removal of non-bacterial sequences.
- Combine figures in multi-panels. Why you not consider samples in only one ordination? Same for Alpha-diversity? In your experimental design, you state that you want to compare the effect of mangrove plants (i.e., soil influenced by the plant) vs bulk soil across two locations.
- Proper statistical analysis should support the ordination graphs to confirm diversity among communities.
- Provide figures with high resolution; it is difficult to read some of them.

---

## Round 0.2 · accepted · Accept

I believe that the authors need to pay attention to few details before publishing. However, the manuscript is nearly ready.

Reviewer 1 ·

Basic reporting

no comment

Experimental design

no comment

Validity of the findings

no comment

Additional comments

Line 181: Italicize Rhizophora mucronata.
Lines 312-313: "Three bulk soil samples were obtained from each site in the same way." Clarify: in the same way as what?
Methods section: Add references for the programs used.
Line 422: Remove "metagenomics"